# Numerical Parametric Study of Coda Wave Interferometry Sensitivity to Microcrack Change in a Multiple Scattering Medium

**DOI:** 10.3390/ma15134455

**Published:** 2022-06-24

**Authors:** Bin Ma, Shukui Liu, Zhanguo Ma, Qi-Ang Wang, Zibo Yu

**Affiliations:** 1State Key Laboratory for Geomechanics and Deep Underground Engineering, China University of Mining and Technology, Xuzhou 221116, China; mabincumt@163.com (B.M.); zgma@cumt.edu.cn (Z.M.); qawang@cumt.edu.cn (Q.-A.W.); 2Department of Civil Engineering, The University of Nebraska at Lincoln, Omaha, NE 68182, USA; 3School of Mechanics and Civil Engineering, China University of Mining and Technology, Xuzhou 221116, China; ts20030219p31@cumt.edu.cn

**Keywords:** 3D printing concrete, coda wave interferometry, crack angle and length, multiple scattering media, numerical modeling

## Abstract

The expansion of cracks in 3D printing concrete materials may lead to structural failure, so it is essential to monitor crack propagation development. Coda wave interferometry (CWI) has been proven to be sensitive to microcracks, however, the evolution pattern of ultrasonic coda waves during crack growth is still not clear. This paper reports a numerical study of the sensitivity and feasibility of CWI for monitoring microcrack growth in heterogeneous materials. A two-phase concrete model, which contains microcracks with different angles and lengths, was developed using the finite element analysis software ABAQUS. The relative velocity change (Δv/v) and the decorrelation coefficient (Kd) at different crack increments were quantitatively analyzed. The numerical simulation results show that coda waves are sensitive to microcrack length as well as the crack angle. The Δv/v increases linearly with the increase of the length of a single microcrack, and the Kd could be linked to the crack length quadratically. Furthermore, a quantitative functional relationship between the CWI observations (Kd, Δv/v) and the angle of the crack to the source/receiver and the relative length growth of the crack are established. In addition, the nonlinear relationship between slope and angle can be fitted with a sinusoidal function. The reported results quantitatively assess the coda wave variation pattern during crack propagation, which is important for the promotion and application of CWI technology.

## 1. Introduction

Three-dimensional printing is a rapid prototyping technology that uses three-dimensional digital models and mechanical equipment to form materials in piles [1,2,3]. Three-dimensional printing technology can quickly create geometrically complex structures, significantly reduce industrial waste, production time and labor costs, and is considered to be the core technology of digital manufacturing [4]. Three-dimensional printed concrete will have defects in the production or service process, the existence of these defects will reduce its service life. Therefore, it is particularly important to monitor the structural health of 3D printed concrete materials.

Coda wave interferometry (CWI) is a new method for micro damage identification in civil engineering. Ultrasound propagation in a multiple scattering medium is subject to strong scattering, resulting in the rapid attenuation of ultrasound energy in the medium [5]. Concrete is a common heterogeneous medium with a complex composition, consisting of coarse aggregates and fine sands, which exhibit strong inhomogeneities. Therefore, ultrasonic waves propagating in concrete are strongly scattered. In the field of non-destructive testing (NDT) of concrete, it is typical to increase the ultrasonic frequency to improve the sensitivity, but the attenuation of the direct wave also increases rapidly [6]. The coda wave, which is usually considered as noise, contains a large amount of physical information. In fact, the coda waves that are scattered several times are extremely sensitive to small changes inside complex media. Coda wave interferometry (CWI) is known as a highly sensitive method for small changes in heterogeneous media and has shown significant advantages in laboratory studies in the field of damage growth monitoring [7,8,9,10]. In laboratory experiments, experimental results for monitoring the crack propagation process in materials are influenced by a combination of factors such as external loads [11] and ambient temperature [12]. It is difficult to determine the sensitivity of the CWI technique to crack propagation and the evolution of the coda wave under the simultaneous effects of multiple influencing factors. Numerical simulation experiments have unparalleled advantages over laboratory experiments in terms of variable control. However, the existing numerical simulations of multiple scattering media are still not perfect: the coarse aggregate is not modeled [9] or only the reflection of the aggregate is considered instead of the transmission of the aggregate [13], which does not provide a realistic simulation of the propagation of ultrasonic waves in real media.

Coda waves analysis techniques have been actively studied and reported in recent decades due to the high sensitivity of coda waves in detecting early and small damages. R. Snieder [12] was the first to propose the CWI technique, through which a nonlinear relationship between seismic wave velocity and temperature in granite was determined. Larose et al. [14] constructed a probabilistic mapping model between the coda wave decorrelation coefficient and the damage location inside the medium using a sensitive kernel function based on the theory of multiple scattering wave propagation in heterogeneous media, while this study did not indicate the relationship between the correlation coefficient and the development of internal damage. Zhou et al. [15] proposed a CWI-enabled fatigue cracks detection method using piezoceramic transducers and deduced a linear relationship between crack width and relative velocity change based on the acoustoelastic theory, however, the study was mainly for surface cracks and not yet applicable for internal crack propagation.

The coda waves observed in the laboratory tests are often affected by multiple factors, thus the parametric study of factors that may affect the change of coda waves during the crack propagation process is of significant meaning to understand. For this issue, using the numerical simulation method is a good research idea. Numerical simulations have been extensively studied in the simulation of ultrasonics [16,17,18]. Numerical simulation methods have tremendous advantages over laboratory studies in terms of variable control and greater flexibility to capture both single and multiple scattering [19,20,21]. However, most of these studies are still limited to the feasibility of simulating wave propagation in heterogeneous media. Recently the simulation of ultrasonics in heterogeneous media started to enter the practical application level, such as the simulation of embedded piezoelectric transducers for the monitoring of concrete cracks [22,23]. Chen et al. [13,24] used the spectral element method to elaborate the phenomenon of nonlinear wave interferometry in two-dimensional heterogeneous media and investigated the sensitivity of the CWI technique to crack propagation. Similar to Chen’s work, many scholars conducted a large number of numerical simulation experiments and laboratory tests from a macro perspective to study the sensitivity of the coda interference method to microcrack changes [8,25,26]. However, the scattering problem during the growth of a single crack in the heterogeneous medium is still not fully understood. Chen et al. studied the influence of receiver source position on the observed value of the coda by changing the receiver position, but this method ignored the influence of the distance from the emitter to the damaged area when changing the position [6].

Before studying the propagation process of ultrasonic waves in concrete, it is necessary to understand the properties of the concrete medium. Concrete is a heterogeneous multiphase composite condensate composed of hardened cement paste, sand, gravel, and other aggregates. In simple terms, concrete can be considered as a two-phase composite composed of cement mortar matrix and aggregate fillers. Ultrasound propagation in concrete is bound to be affected by heterogeneous scatterers such as aggregates. Therefore, it is crucial to establish a concrete model to simulate the wave propagation process more practically. This paper aims to deepen the understanding of the scattering wave propagation rule during damage development in heterogeneous media using numerical simulations.

In this paper, a two-dimensional heterogeneous concrete model with damage, which is set to be cracks with different angles and lengths, was developed, and the evolution of the coda wave during crack propagation was observed. The relationship between the relative velocity change and decorrelation coefficient during the expansion of microcracks and the crack expansion was quantified. In addition, the effect of crack angles was analyzed. The arrangement of this paper is as follows: In Section 2, the theoretical background involved in this study is introduced. Section 3 details the configuration of the model and the selection of parameters. Section 4 presents the analysis and discussion of the numerical simulation results. Specific conclusions are given in Section 5.

## 2. Theoretical Background

### 2.1. The CWI Theory

CWI is a technique that detects small changes in a medium by using multiple scattering waves, allowing the determination of the relative velocity change of a diffuse wavefield measured from a fixed source and a fixed receiver at two different moments [27]. Ultrasonic waves are scattered many times by cracks or scatterers inside the medium, and the coda wave after being scattered multiple times is more sensitive to small changes compared to the first arrival wave [28]. Cracks or scatterers produce small changes that are amplified many times, making coda wave analysis an excellent tool for observing minor damage.

According to the path summation theory of CWI, the waveform in the reference wavefield $u\left(t\right)$ can be expressed as a sum of the waves propagating along all possible paths [29], the reference wavefield $u\left(t\right)$ is given by
(1)$$u\left(t\right)=\underset{p}{\sum }S_{p}\left(t\right)$$
where p represents the travel path, $S_{p}$ represents the wave corresponding to the path, t is the travel time along the path. When the size of crack propagation is much smaller than the mean free path, the dominant effect on the waveform arises from the change in the travel time $\tau_{p}$ of the wave that travels along each path. The perturbed wavefield $u_{p}\left(t\right)$ is given by
(2)$$u_{p}\left(t\right)=\underset{p}{\sum }S_{p}\left(t-\tau_{p}\right)$$

CWI can be used to assess the correlation and relative velocity change of two signals. The stretching method is a powerful tool for estimating the difference between signals, which calculates the correlation $CC\left(\varepsilon \right)$ between the reference wavefield $u\left(t\right)$ and the perturbed wavefield $u_{p}\left(t\right)$. The cross-correlation function $CC\left(\varepsilon \right)$ is given by
(3)$$CC\left(\varepsilon \right)=\frac{\int_{t_{c}-T}^{t_{c}+T}u\left(t\left(1+\varepsilon \right)\right)u_{p}\left(t\right)dt}{\sqrt{\int_{t_{c}-T}^{t_{c}+T}u^{2}\left(t\left(1+\varepsilon \right)\right)dt\int_{t_{c}-T}^{t_{c}+T}u_{p}^{2}\left(t\right)dt}}$$

$CC\left(\varepsilon \right)$ quantitatively indicates the similarity of the two signals before and after disturbance of the medium within a given time window $\left[t_{c}+T,t_{c}-T\right]$, where $t_{c}$ is the central time of the window and 2T is the length of the selected window frame. The *ε* that makes the maximum value of CC is equal to the relative velocity change, which can be written as $\varepsilon_{max}=-\Delta v/v$. Furthermore, in order to determine the degree of distortion between the two signals, a decorrelation coefficient is introduced as $Kd=1-CC\left(\varepsilon \right)$. Kd is used to find single [14] or multiple local scatterer disturbances in the medium [30].

### 2.2. Applying CWI to Monitoring Changes in Scattering Media

The total scattering coefficient $g_{0}$ [31] can be used to define the scattering power of a medium. Assume an ideal scattering medium consists of a random distribution of n point-like scatterers, with a number density $\rho_{n}$, within a background velocity $V_{0}$, and total scattering cross section $\sigma_{T}$, this coefficient can be defined as $g_{0}=\rho_{n}\sigma_{T}$. Rossetto et al. [32] used diffusion propagation theory to derive expressions related to the theoretical solution caused by local perturbations of $\sigma_{T}$. The knowledge of the sensitivity kernel $Q\left(s,x_{m},r,t\right)$ needs to be introduced first. Q denotes the probability that the wave is emitted at source s, passes through position $x_{m}$, and is received at receiver r after a total propagation time t. This probability describes the spatial density of the coda wave at the position. $K_{d}$ was rewritten according to the change in the total scattering coefficient $g_{0}$ of the material within the background velocity $V_{0}$ as,
(4)$$Kd\left(X,t\right)=\frac{V_{0}}{2}\int_{V_{K}}\left|\Delta g_{0_{p-u}}\left(x\right)\right|Q\left(s,x,r,t\right)dV_{Q}\left(x\right)$$
where *X* defines the set of n randomly distributed perturbation locations within the inner kernel volume $V_{Q}$, and the difference in the total scattering coefficient between the perturbed and unperturbed material states is defined by $\left|\Delta g_{0_{p-u}}\right|$. In terms of Equation (4), any change in the contrast of the impedance and the size of the scatterer leads to an increase in the decorrelation. In addition, these changes will alter the strength of the diffusive wavefield through the diffusion and attenuation coefficients of the material [33,34] leading to a change in the sensitivity kernel.

### 2.3. Stress Wave Simulation Based on Finite Element Method

The simulation of scattering waves is a stress wave propagation problem that should be dealt with using a dynamic solver. In this paper, the scattering waves are simulated using the ABAQUS/Explicit solver [35]. Stress waves propagation at finite elements is studied using a governing equation and an equilibrium equation for each element and node depending on the time step, as follows:(5)$$\rho \ddot{u}+\frac{\partial \sigma }{\partial x}=0$$
where ρ is the mass density, $\ddot{u}$ is the particle acceleration, x is the position and σ is the stress, and
(6)$$\rho \ddot{u_{i}}=G\frac{\partial^{2}u_{i}}{\partial x_{j}\partial x_{j}}+\left(\lambda +G\right)\frac{\partial^{2}u_{j}}{\partial x_{i}\partial x_{j}}$$
where u is the particle displacement, i and *j* are finite element position indices, λ is the Lame’s constant of the medium, and G is the medium’s shear modulus.

## 3. Modeling and Numerical Simulation

In this paper, concrete was simulated by considering it as a two-phase heterogeneous medium composed of coarse aggregate and cement mortar. The heterogeneity in the concrete causes a strong scattering of ultrasonic waves in the wavelength range of the aggregates. To make the scattering waves in concrete more realistic, concrete aggregates were materialized by multiple randomly distributed circular areas with different material properties, and local damage was achieved by inclusions with high aspect ratios. The technical route adopted in this paper is shown in the flowchart in Figure 1.

Figure 2 shows a series of rectangular finite element models with a width of 400 mm and a height of 400 mm. Walraven [36] derived the probability of aggregate distribution for an arbitrary particle size on a two-dimensional cross-section. Based on this theory, a Python script was used to control the particle size and position of the coarse aggregates. The mechanical properties of mortar and aggregates are reported in Table 1. A circular area of 80 mm in diameter, which has the same material properties as the mortar in the model, was placed in the center of the model. The initial crack was an ellipse with a long axis of 20 mm and a short axis of 0.05 mm. The growth process of the crack was simulated by increasing the length of the ellipse by ΔL. The angle between the long axis of the crack and the horizontal direction is θ, and the crack propagation and the change of angle θ are schematically listed in Figure 3. Eleven crack length configurations were created for each damage level with the different values of $L_{n}$ ($L_{n}=L+\Delta L.\Delta L=nL/100\;;n=0,1,\;2\ldots \;10$). Each damage level ($L_{n}$) has seven angle variations, which are controlled by θ (θ = 0°, 15°… 90°). In order to maximize the acoustic impedance coefficient and increase the reflection coefficient, the interior of the crack was set to void [13]. The model boundary was considered to be rigid, where the stress waves are entirely reflected.

In the numerical simulation of elastic waves, especially in the field of seismic exploration, the source signal is mainly simulated using a Reckonic or Gaussian function and its first-order derivative; while in the numerical simulation of high-frequency ultrasonic waves, the excitation source signal often uses a convex cosine function to resemble more closely the impulse signal from the ultrasonic transducer, as in Equation (7) [37]:(7)$$f\left(t;n\right)=\left\{\begin{array}{ll} \frac{{\left(-1\right)}^{n}}{2}\left[1-cos\left(\frac{\omega_{0}}{n}t\right)\right]cos\left(\omega_{0}t\right), & 0\leq t\leq n\frac{2\pi }{\omega_{0}}\\ 0, & t>n\frac{2\pi }{\omega_{0}} \end{array}\right.$$
where $\omega_{0}=2\pi f$, *f* is the center frequency, *n* indicates the signal duration period, the larger the value of n, the narrower the frequency band, which is conducive to reducing the dispersion effect. The signal emission source was placed at (20 mm, 40 mm) to simulate the ultrasonic waves emitted by the piezoelectric transducer. The acceleration within 0*–*4 ms at (20 mm, 0 mm) was extracted as the receiving scattered signal. The ultrasound source and mesh division are shown in Figure 4.

Finite element simulation of the stress wave propagation process was performed using an explicit solver. The explicit method achieves satisfactory accuracy provided that the time increments are small enough, and it is very effective in solving for a large number of time increments, which are typically upwards of 10^5^ steps [38]. The explicit computational procedure uses central difference operators and small time increments for integration, which are required to satisfy stable time increments Δt, the time increments must be fewer than the critical one:(8)$$\Delta t=min\left(\frac{l_{e}}{V}\right)$$
where $l_{e}$ is the smallest size of the finite element and V is the wave velocity.

The number of finite elements per wavelength is an essential factor while modeling wave propagation with the finite element method. According to Drozdz’s study [39], each wavelength should cover at least seven finite elements to ensure the accuracy of the simulation results. In this paper, the minimum wavelength is 14 mm, and the minimum wavelength covering seven elements was finally adopted under the comprehensive consideration of computational efficiency and simulation accuracy. As shown in Figure 4c, the concrete and coarse aggregate mesh are both 2 mm, while the crack region is meshed more finely for cell division to fit the size of the crack. The interior of the crack is considered to be empty, so no meshing is performed. The mesh size directly affects the minimum time increment, and the reduction of the finite element size leads to a significant increase in computational complexity. Considering the minimum finite element size and computational efficiency, the minimum time increment is set to 10^−8^ s. Table 2 lists the mesh size and time increments.

## 4. Results and Discussion

### 4.1. Dependence of CWI Observations on the Variation of Crack Length

In total, 77 configurations with different crack angles and lengths were numerically investigated. The variation of lengths indicates the process of crack expansion. Figure 5 shows a snapshot of the wavefield in a homogeneous medium, the stress waves show a uniform diffusion, and the wavefield intensity is similar at the same distance from the emitting source [40,41]. Compared with the wavefield in a homogeneous medium, the residual vibrations always exist in the places covered by ultrasonic waves. Due to the different material parameters of the aggregate and cement mortar matrix, the ultrasonic waves are constantly reflected and refracted between the elastic partition interfaces, resulting in multiple scattering phenomena. Furthermore, the more scatterers in the medium, the slower the coda wave attenuation rate [32,42], which eventually leads to small residual vibrations of ultrasonic waves. This phenomenon also proves the reliability of the finite element method in simulating the propagation of ultrasonic waves in solid heterogeneous media.

The exemplary waveforms of θ=45° are shown in Figure 6, which match well with the waveforms obtained from concrete testing in the laboratory [43,44,45], this also implies that the simulation method adopted in this paper is convincing. The overall amplitude and phase of the signal exhibit a high level of coherence, and it can be seen that the amplitude and phase of the first arrival wave are almost identical, even with different lengths of cracks. However, as the crack grows longer, a significant time shift could be observed in the coda wave, indicating that the time shift formed by multiple scattering contains rich information that cannot be reflected by the first arrival wave.

The stretching method was applied to all 77 signals, and the selection of the time window in the calculation process of this method is crucial. The longer the time window, the higher the cost of CWI calculation, the relative velocity change tends to stabilize when the window centroid is sufficiently late, improving the robustness of CWI [46,47,48]. A compromise between analytical robustness and the computational cost was chosen for the time window of [1.5 ms, 2 ms]. For each model with different crack angle θ, the waveform of $L_{0}$ was used as the reference signal to calculate the cross-correlation coefficient between the waveforms of other lengths. Then, Δv/v and Kd were obtained from the cross-correlation coefficient. Figure 7 shows the correspondence between the Δv/v and the crack length growth, and the dots represent the Δv/v during the crack propagation. It can be seen that for each angle, the increase in crack length leads to a decrease in the relative velocity, and a strong correlation could be observed. The solid line shows the best linear fit regression curve, $R^{2}$ indicates the closeness of the fit, and the linear correlation between Δv/v and ΔL/L is evident for each group. All fitted curves can be represented uniformly by Equation (9), where $\alpha_{\delta v/v}^{\theta }$ denotes the slope at θ.
(9)$$\Delta v/v=\alpha_{\delta v/v}^{\theta }\cdot \frac{\Delta L}{L}$$

Figure 8 shows the relationship between the Kd
and the crack length growth, and the dots indicate the variation of the Kd with the crack length, while the solid line is the best quadratic regression fit. In contrast to Δv/v, Kd is no longer linearly correlated with ΔL/L. For each fitted curve $R^{2}>$ 0.94, which indicates that the results of this quadratic regression are satisfactory. All fitted curves can be represented uniformly by Equation (10), where $\alpha_{Kd}^{\theta }$ denotes the slope at θ.
(10)$$Kd=\alpha_{Kd}^{\theta }\cdot {\left(\frac{\Delta L}{L}\right)}^{2}$$

The propagation path of the incident wave into the interior of the medium is affected by the scatterer, resulting in a change in propagation direction and redistribution of energy. The change in the scatterer cross-section causes a complex path transition inside, resulting in a change in the Δv/v and Kd. According to the results of numerical experiments, it can be assumed that an increase in the scatterer cross-section length leads to a linear decrease in the Δv/v. The size of the scatterer affects the scattering coefficient $g_{0}$, the background velocity $V_{0}$, and the sensitivity kernel Q. According to Equation (4), the Kd is influenced by the combination of $g_{0}$, $V_{0}$ and Q. Therefore, the size change of the scatterer leads to a complex chain reaction inside, which eventually only allows the approximate relationship between the change of the crack length and the Kd to be observed macroscopically.

### 4.2. Influence of Relative Position of Source/Receiver and Cracks on CWI Observations

Figure 9 shows the slope of CWI observations with crack length versus crack angle θ (0–π/2), a trigonometric function was used to fit the trend of the slope. The relationship between $\alpha_{\Delta v/v}^{\theta }$ and $\alpha_{Kd}^{\theta }$ with crack length growth and angle change is obtained by combining Equations (9) and (10), see Equations (11) and (12).
(11)$$\Delta v/v=(y_{0}+A_{0}sin\left(\frac{\pi \left(\theta -\theta_{0}\right)}{\omega_{0}}\right))\cdot \frac{\Delta L}{L}$$
(12)$$Kd=(y_{1}+A_{1}sin\left(\frac{\pi \left(\theta -\theta_{1}\right)}{\omega_{1}}\right))\cdot {\left(\frac{\Delta L}{L}\right)}^{2}$$

According to Equations (11) and (12), the values of Δv/v and Kd can be uniquely determined for any crack length at any angle θ (0–π/2). In addition, it can be seen that both $\alpha_{\Delta v/v}^{\theta }$ and $\alpha_{Kd}^{\theta }$ fluctuate up and down around a constant, which indicates that for single crack growth monitoring, the location of the source/receiver can affect accuracy, but crack growth can be detected anywhere around the damage. The reason for this phenomenon may be that the change of microcrack angle and length changes the local coda wave path, and then changes the observations. Note that this relationship is applicable to a single crack, for more realistic cases, like concrete with multiple cracks with different lengths and orientations, this law may not apply. The problem is complicated when there are multiple cracks in the damage zone, and the study of a single crack can improve our understanding of the coda wave evolution of the crack expansion growth process. After multiple interactions of a single crack with scattered waves, the angle is no longer necessary for monitoring, so it is reasonable to assume that when multiple angular cracks are present in an area at the same time, or when damage occurs in a local area, the phase delay due to these damages is approximately the same when detected anywhere around. In more visual terms, the relative position of the source/receiver has little effect on the identification of damage when length growth occurs for multiple random angle cracks. Similar conclusions were obtained by Chen et al. [6] in their experiments to observe the CWI observations by changing the relative positions of the source and receiver.

## 5. Conclusions

In this paper, a two-dimensional heterogeneous medium has been modeled by the finite element method, and the quantitative assessment of material damage by the CWI technique is investigated. The variation of Δv/v and Kd during the crack extension was studied by changing the crack length to simulate the crack propagation process. The results show that Δv/v is linearly related to the length growth rate of a single crack, and Kd is linearly related to the square of the length growth rate of a single crack. Furthermore, the crack extension can be recognized under different angles, but the recognition effect is different under different angles. The nonlinear relationship between slope and angle can be fitted with a sinusoidal function, making it possible to predict the CWI observations at different angles and different length increments.

This paper reveals the change rule of Δv/v and Kd during the expansion of a single crack and provides a theoretical basis for the placement of sensors in practical testing, which promotes the promotion and application of CWI technology in nondestructive testing. However, the reported model does not fully simulate the actual situation of ultrasonic waves in concrete because the shape of coarse aggregate and the transition zone of the aggregate interface are ignored. In addition, the simulation experiments for multiple cracks acting simultaneously were not performed. In future experiments, we intend to refine the model details and consider multiple microcracks and different aggregate diameters to make the conclusions more systematic.

## Figures and Tables

**Figure 1 materials-15-04455-f001:**
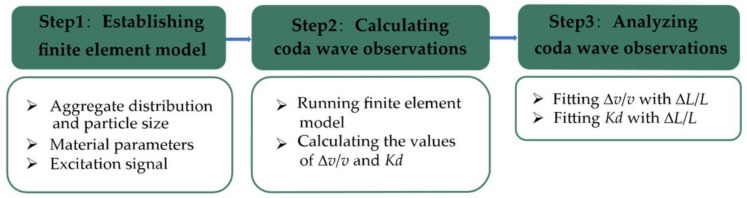
Flowchart of coda wave evolution in concrete using numerical simulation.

**Figure 2 materials-15-04455-f002:**
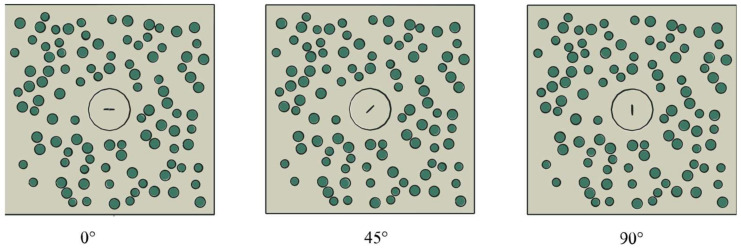
Numerical configuration of a damaged concrete model with random aggregates. The model size was 400 mm × 400 mm. Cracks at different angles to the horizontal were located in a circular damage zone centered at (200 mm, 200 mm). The size of the crack was initially set to be 20 mm × 0.05 mm.

**Figure 3 materials-15-04455-f003:**
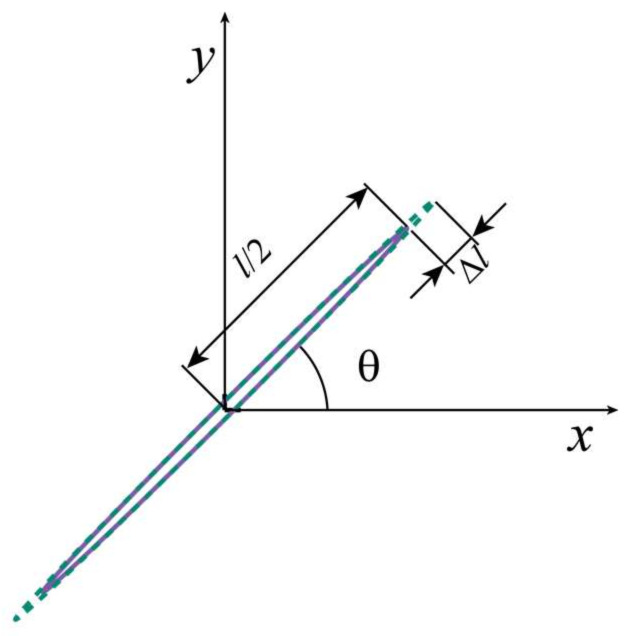
Schematic diagram of crack propagation. The solid line indicates the initial crack, and the dashed line indicates the crack after development. The angle θ increases by a 15° increment from 0° to 90°.

**Figure 4 materials-15-04455-f004:**
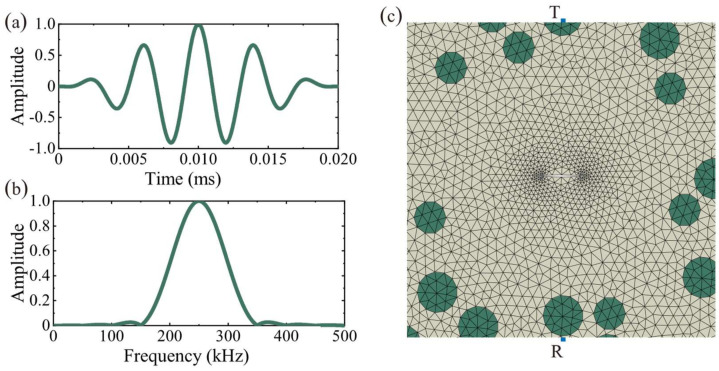
Numerical simulation configuration: (**a**) Source signal in the time domain signals (normalized amplitude); (**b**) Source signal in the frequency domain (normalized amplitude); (**c**) Mesh division and piezoelectric transducer distribution (T: Transmitter, R: Receiver).

**Figure 5 materials-15-04455-f005:**
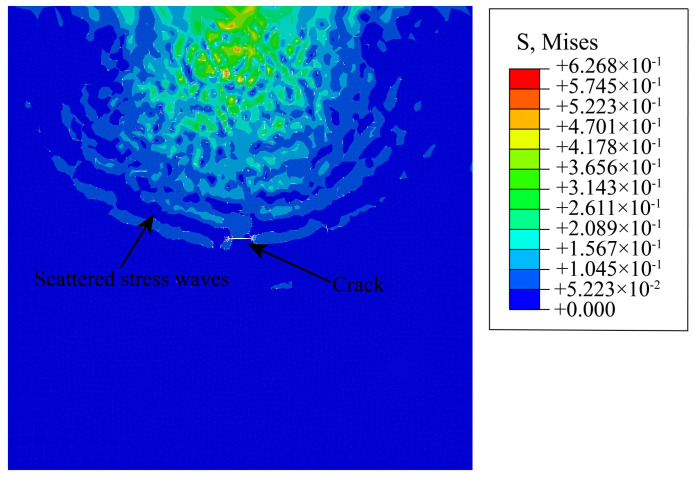
Wavefield of the model with a crack simulated with ABAQUS at 0.1 ms (θ=0°, L=2 mm).

**Figure 6 materials-15-04455-f006:**
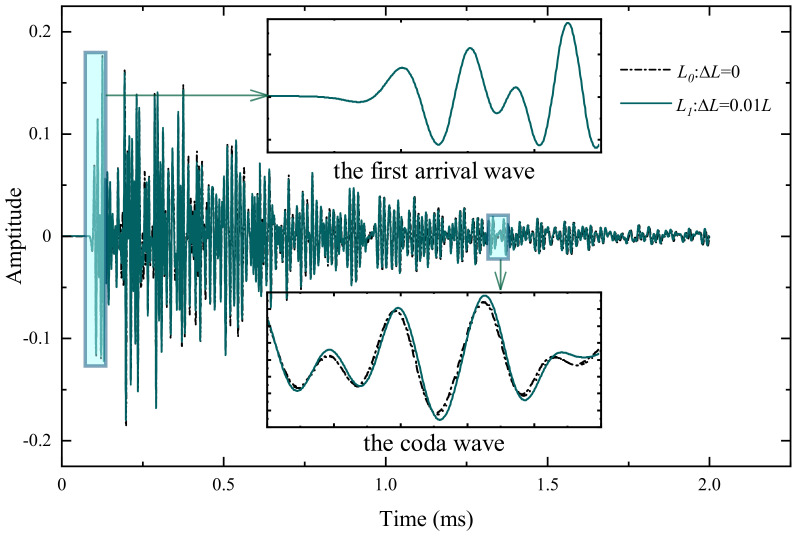
Examples of two records acquired at the same location (same transducer but on two different lengths of cracks). Between each acquisition, the crack length increases by 0.01 L, which is barely visible in the early part (inset between 0.08 and 0.13 ms) of the record, but very visible in the later part (inset between 1.55 and 1.60 ms).

**Figure 7 materials-15-04455-f007:**
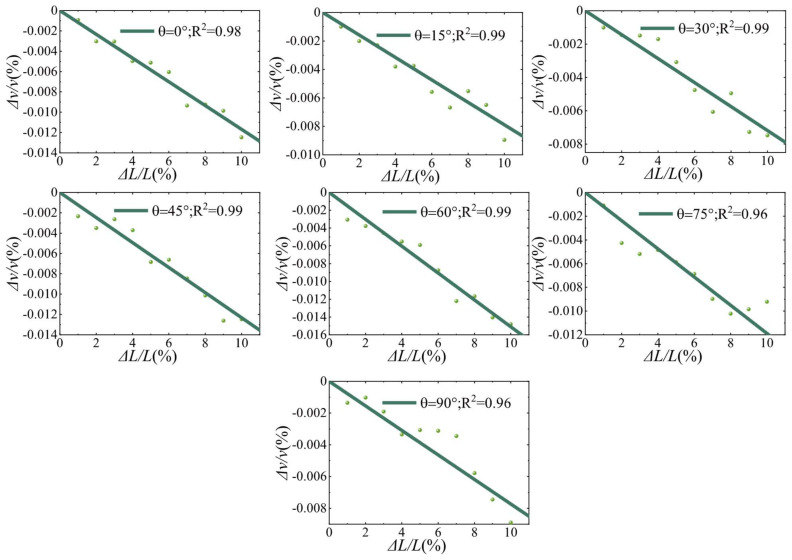
Correspondence between observed values of relative velocity change (Δv/v) and crack length change. For each model with an angle θ, eleven numerical models with different crack length were used.

**Figure 8 materials-15-04455-f008:**
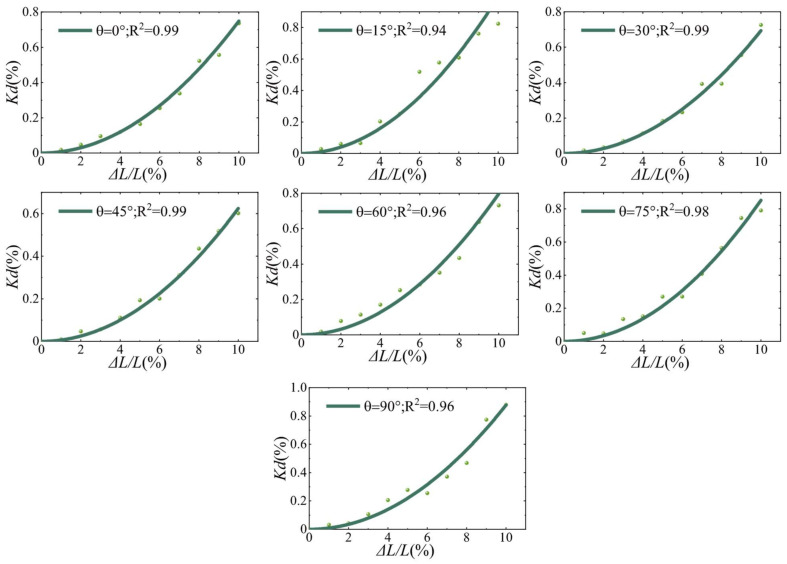
Correspondence between observed values of decorrelation coefficient (Kd) and crack length change. For each model with an angle θ, eleven numerical models with different crack lengths were used.

**Figure 9 materials-15-04455-f009:**
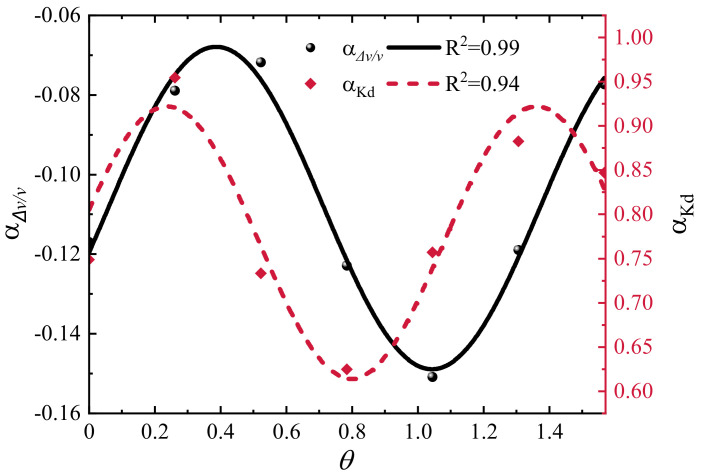
Slopes of CWI observations with crack length changes versus θ.

**Table 1 materials-15-04455-t001:** Properties of the materials used in the numerical simulation model.

Material	Mass Dencity $\left(\rho \right)\left(\mathrm{kg}/m^{3}\right)$	Young’s Modulus (E) (GPa)	Poison’s Ratio $\left(\nu \right)$	P-Wave Velocity $(V_{p})\;\left(m/s\right)$	S-Wave Velocity $(V_{s})\;\left(m/s\right)$
Mortar	2000	30	0.2	4082	2500
Aggregate	2400	60	0.2	5270	3227

**Table 2 materials-15-04455-t002:** Mesh partition setting.

The Shortest Wavelength	14 mm
Element size	2 mm
Number of elements per wavelength	7
Δt	1 × 10^−8^ s

## Data Availability

Not applicable.

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
