# Peer review of "Numerical Parametric Study of Coda Wave Interferometry Sensitivity to Microcrack Change in a Multiple Scattering Medium"

_materials, 2022, doi:10.3390/ma15134455_

Round 1
Reviewer 1 Report
The paper concerns the current issues of assessing the defects of 3D printing concrete in relation to, inter alia, its viability. Monitoring the structure of 3D printing concrete with the use of Coda wave interferometry (CWI) is an important issue that contributes to the development of the process of assessing the correctness of the structure of 3D printing concrete. The issues presented in the paper contribute to the evaluation of the practical numerical side of simulation of the wave propagation process as a result of the development of internal damage the structure of the concrete which is heterogeneous multiphase composite condensate composed of hardened cement paste, sand, gravel, and other aggregates.
Remarks:
- There is no axle designation in Figure 2.
- Is there a screen from ABACUS in Figure 4? Please add this information in the figure caption.
Reviewer 2 Report
The work is theoretical in nature, but it testifies to the great scientific experience of the authors. The following issues require clarification: 1) what material parameters the concrete adopted for modeling was characterized by, 2) on what basis and the concrete for modeling was selected, 3) whether in the modeled transition zone between aggregate grains and cement slurry, 4) or in the model the model variable strength of the concrete mix along the section height, 5) whether calculations were made for a different grain diameter and other strengths. The work requires supplementing with the content of the formulated questions and doubts
Reviewer 3 Report
The authors present an interesting study on the sensitivity of Coda wave interferometry in detecting the growth of microcracks in cement. To this aim, they realized detailed numerical models. In each model, the concrete was reported as two-phase material and the cracks were considered with different angles and lengths.
The paper is clear, and the main ideas are well developed. Nevertheless, some clarifications are required to improve the quality of the paper before publication.
-Please use the italic text in the main text when recalling the formulas' parameters for example Kd, u(t), etc..
-Please check the phrase from 72 to 77 lines. There is an extra dot in line 76.
-On page 3 line 125 is the travel path "p" or "Sp"?
-On page 4 line 161 in the phrase " be dealt with by a dynamic solver" " delete "by" or "with".
-Please, in table 1 correct the parenthesis position of Vs
-Please, add the legend to figure 4 and change the text color from red to black or white
Reviewer 4 Report
The paper "Numerical parametric study of Coda Wave Interferometry sensitivity to microcrack angle and length in a multiple scattering medium" is interesting and may be published after necessary corrections:
(1) The title is confusing to readers, it should be reformulated.;
(2) The abstract needs more quantitative information on the results found;
(3) The authors can provide more evidence at the end of the introduction of the innovation and final quality of this paper compared to other studies already published;
(4) A methodological flowchart must be inserted;
(5) The results presented in Figure 8 should be further discussed by the authors;
(6) The conclusion needs to be better explained, there are still points of doubt in the final document.
